# Inhibition of oxidative stress in cholinergic projection neurons fully rescues aging-associated olfactory circuit degeneration in *Drosophila*

Ashiq Hussain[1†], Atefeh Pooryasin[2,3†], Mo Zhang[4†], Laura F Loschek[4], Marco La Fortezza[5], Anja B Friedrich[1], Catherine-Marie Blais[1], Habibe K Üçpunar[4], Vicente A Yépez[6], Martin Lehmann[7], Nicolas Gompel[5], Julien Gagneur[6], Stephan J Sigrist[2,3], Ilona C Grunwald Kadow[1,4,8*]

[1]TUM School of Life Sciences, Technical University of Munich, Freising, Germany; [2]Institute of Biology, Free University of Berlin, Neurogenetics, Germany; [3]Cluster of Excellence NeuroCure, Charité-Universitätsmedizin Berlin, Berlin, Germany; [4]Max-Planck Institute of Neurobiology, Martinsried, Germany; [5]Fakultät für Biologie, Biozentrum, Ludwig-Maximilians-Universität München, München, Germany; [6]Department of Informatics, Technical University of Munich, Garching, Germany; [7]Department of Molecular Pharmacology and Cell Biology, Leibniz-Forschungsinstitut für Molekulare Pharmakologie, Berlin, Germany; [8]ZIEL – Institute for Food and Health, Technical University of Munich, Freising, Germany

*For correspondence:
ilona.grunwald@tum.de

†These authors contributed equally to this work

Competing interests: The authors declare that no competing interests exist.

**Abstract** Loss of the sense of smell is among the first signs of natural aging and neurodegenerative diseases such as Alzheimer's and Parkinson's. Cellular and molecular mechanisms promoting this smell loss are not understood. Here, we show that *Drosophila melanogaster* also loses olfaction before vision with age. Within the olfactory circuit, cholinergic projection neurons show a reduced odor response accompanied by a defect in axonal integrity and reduction in synaptic marker proteins. Using behavioral functional screening, we pinpoint that expression of the mitochondrial reactive oxygen scavenger SOD2 in cholinergic projection neurons is necessary and sufficient to prevent smell degeneration in aging flies. Together, our data suggest that oxidative stress induced axonal degeneration in a single class of neurons drives the functional decline of an entire neural network and the behavior it controls. Given the important role of the cholinergic system in neurodegeneration, the fly olfactory system could be a useful model for the identification of drug targets.
DOI: https://doi.org/10.7554/eLife.32018.001

In order to exploit *Drosophila* to characterize the mechanisms of neural circuit degeneration, we first established that flies show a similar early aging-dependent decline in olfactory perception as humans (*Doty et al., 1984*). In olfactory T-maze assays, where the animal's preference or aversion for an odor is recorded as an index of their approach or avoidance behavior toward the odorant, we found that the performance to eight different (three attractive, five aversive) odors gradually declined with age (*Figure 1A* and *Figure 1—figure supplement 1A–D*). This decline occurred also for behavior to odors that are detected independent of the canonical olfactory receptor ORCO (olfactory receptor co-receptor), and hence affected all tested odorants recognized by the three classes of olfactory receptors (ORs, ionotropic receptors (IRs), and gustatory receptors (GRs); *Figure 1—figure supplement 1E–I*). By contrast, the fly's high attraction to blue light was not significantly different between

**Figure 1.** The sense of smell ages faster than the sense of vision. (**A**) Olfactory preference index of aging Canton S flies to aversive (benzaldehyde, 3-octanol) and attractive odors (2,3-butanedione, putrescine) in the T-maze assay. There is a gradual and significant (p≤0.01) decrease in olfactory preference with aging (1–10 weeks). For additional odors see *Figure 1—figure supplement 1*. (**B**) Preference index (X-axis) of flies to blue light versus red light in the T-maze assay against age (Y-axis). There was no significant difference between the data points. Graphs show mean value ±SEM (n = 8 trials, 60 flies/trial 30 ♀ and 30 ♂). (**C**) Schematic illustration of the fly brain and antennal appendages with olfactory sensory neurons (OSNs). OSNs project into the antennal lobe (AL), where they innervate a specific glomerulus (green). Projection neurons (PNs, blue) send the information mainly to two higher brain centres, the mushroom body (MB) and the lateral horn (LH) (top).

DOI: https://doi.org/10.7554/eLife.32018.002

The following source data and figure supplement are available for figure 1:

**Source data 1.** Source data for *Figure 1* and *Figure 1—figure supplement 1*.
DOI: https://doi.org/10.7554/eLife.32018.004
**Figure supplement 1.** Functional decline affects odors detected by all receptor classes.
DOI: https://doi.org/10.7554/eLife.32018.003

1 and 10 weeks of age (*Figure 1B*) indicating that the flies were healthy enough to move and to make decisions in this type of behavioral assay. These data argue that in flies, as observed in humans, the sense of smell declines before and/or faster than the sense of vision.

We next sought to identify the underpinning neuronal and genetic mechanisms. First, we asked whether all neurons were affected equally or whether perhaps particular neuron types within the olfactory circuit were primary targets of aging. The architecture of the fly olfactory system parallels that of vertebrates (*Wilson, 2013*). Olfactory sensory neuron (OSN) axons connect to the antennal lobe (AL), the equivalent of the olfactory bulb (OB), in the brain, where olfactory information is further processed and transferred by second order projection neurons (PNs; analogs of mitral and tufted cells) to higher brain centers (*Figure 1C*). A straightforward reason for a loss of sense of smell could be a loss of OSNs or a reduction in their odor responsiveness, for instance through diminished olfactory receptor expression. Yet, by counting the number of a subclass of OSNs we found no difference in the number of sensory neurons between young and older flies (e.g., 1 week vs. 10 weeks *Or42b-Gal4; UAS-mCD8GFP* flies; *Figure 2—figure supplement 1A*), suggesting that OSNs do not die during aging. Furthermore, we detected no significant difference in the size of OSN cell bodies between young and old flies (*Figure 2—figure supplement 1B*). In agreement with these data, RNA-sequencing (RNA-seq) of whole antennae demonstrated that olfactory receptor expression including the expression of those responding to the tested odors and the obligatory co-receptor ORCO, which is required for the detection of the majority of odors, was not significantly different between young and old flies (*Figure 2—figure supplement 1C,D*).

We next investigated possible changes in the OSN's ability to respond to an odor, by recording spikes of neuronal activity in single sensillum recordings, where an electrode is inserted into individual olfactory sensilla (SSR; *Figure 2—figure supplement 2A*). While the response to two aversive

odors, benzaldehyde and $CO_2$ showed a significant decrease in the number of elicited spikes, none of the other recordings using other odorants (six odors) showed a similar decline (*Figure 2—figure supplement 2B–H*). This result did not change even when flies were first sorted by their response to the odor in the T-maze behavioral assay. Flies that reacted to the odorant and flies that appeared to not care showed similar SSR responses, which were also not significantly different between the age groups (*Figure 2—figure supplement 2J,K*). Given that the behavior to odors without a decline in SSR was similarly affected as the behavior to benzaldehyde and $CO_2$, we concluded that a decrease in OSN sensitivity is unlikely to be the general reason for the observed ageing-associated decline in olfactory behavior. Altogether these experiments suggested that a degeneration of peripheral sensory neurons is not the primary reason for the observed ageing-associated phenotype.

OSNs transmit their information to PNs in the central brain (~30 OSNs to one PN; *Figure 1C*). The majority of PNs are cholinergic and can be labeled with the transgenic reporter line GH146-Gal4 (*Figure 2A*). To test whether PNs were functionally impaired, we used *in vivo* calcium imaging (*GH146-Gal4; UAS-GCaMP3.0*) as a proxy of neuronal activity. We first recorded fluorescence changes at the level of the AL, where the dendrites of these neurons receive input from the OSNs, using epifluorescence microscopy (*Figure 2B*). The PN odor response in the responsive glomeruli was significantly reduced at different concentrations including those used in the behavioral assays (1 mM, *Figure 2B*). With the exception of very few odors such as $CO_2$, most odors activate multiple glomeruli. To better distinguish glomerulus-specific responses, we turned to two-photon microscopy and imaged GCaMP fluorescence changes at three different levels of the AL (*Figure 2C,D*). Here, we observed that not all glomeruli were equally affected. While for each odor, one, normally very responsive glomerulus showed a strong and significant decline, two other glomeruli were not significantly affected by age (*Figure 2C,C',D,D'*). The reason for why different glomeruli are differentially affected is not know at this point. Differences in OSN number or olfactory receptor expression can, according to the data shown above, likely be excluded. Currently, we can only speculate that the degree of innervation by inhibitory or excitatory local neurons in the AL, other distinctive features, the activity of the OSN-PN synapses, interaction between sister PNs, or even the interaction with glia cells might play a role in these differences. Nonetheless, the current data strongly suggest that, in contrast to OSNs, PNs are less activated by odors in old as compared to young flies.

PNs transfer their information mainly to two higher brain centers, the MB calyx and the LH. To characterize PN function further, we also imaged GCaMP fluorescence changes at the level of PN boutons in the calyx (*Figure 2E–G*). Remarkably, here the response per PN bouton was not significantly weaker between flies of different ages (*Figure 2F*). However, the number of responsive boutons was strongly reduced in old (4, 6 weeks) as compared to young flies (1 week; *Figure 2G*. Therefore, PN function is affected by aging. Importantly, not only their dendritic response is reduced in a glomerulus-specific manner, their axonal output to higher brain centers drastically diminishes with age. The strength of this decline indicates that primarily axons and their synaptic output regions might degenerate functionally or anatomically due to aging-related mechanisms.

These cellular data suggested that flies suffer from reduced odor sensitivity possibly due to a decline of PN function and a subsequent reduction in information that arrives at higher brain centers. Hence, we wondered whether the animals were still capable of odor recognition and valuation, if compensated for the loss in sensitivity. Indeed, a 10-fold increase in odor concentration was sufficient to markedly improve the old flies' olfactory choice behavior indicating that old flies albeit a decline in odor sensitivity still recognize and correctly valuate a given odor (*Figure 2—figure supplement 3*).

To unravel the mechanism of this functional decline, we analyzed PN morphology in more detail (*Figure 3A*). In contrast to OSNs, the number of PNs labeled by the reporter GH146 >mCD8-GFP was mildly but significantly reduced in old as compared to younger flies (*Figure 3B*). In addition to the mild loss of labeled PNs, their cell bodies had shrunk in older animals as compared to younger animals (*Figure 3C*). Similar observations were made for other aging neurons including mitral cells in aged humans (*Sama-ul-Haq et al., 2008*). Importantly, the pan-neuronally expressed gene *elav* was somewhat upregulated relative to other genes in older brains as compared to its relative expression in younger brains (*Figure 3F*) suggesting that neurons were not selectively lost as compared to other cell types (e.g. glia) in aging *Drosophila* brains. Nevertheless, we observed a significant decrease in ToPro staining, which labels cell nuclei in several areas including the AL (*Figure 3G,H*). Despite of this, the expression level of the neuropil marker N-cadherin (Ncad) in the AL and in the region of the

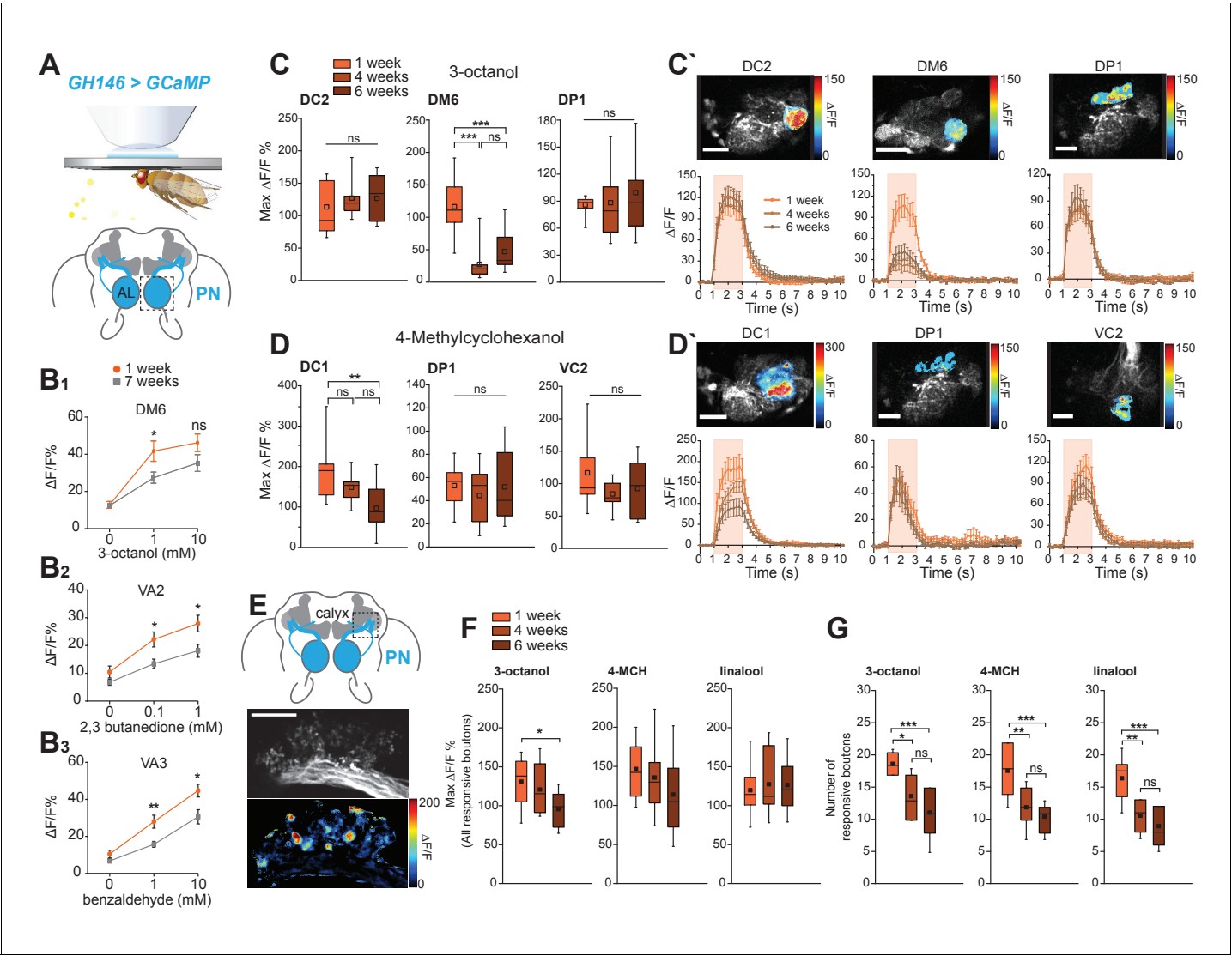

**Figure 2.** Cholinergic projection neurons functionally decay with age. (A) Scheme of in vivo functional imaging preparation. The Ca2+ sensor GCaMP3 is expressed in projection neurons (PNs) under the control of GH146-Gal4 (*GH146-Gal4;UAS-GCaMP3* or GH146 > GCaMP). (B1-3) *In vivo* calcium imaging in PNs (*GH146-Gal4;UAS-GCaMP3.0*) at the level of the AL using epifluorescence microscopy. The neural response of 1 week and 7 weeks old flies to increasing concentrations of benzaldehyde, 2,3-butanedione and 3-octanol was compared (n = 8 ± SEM). Graphs represent the quantification of neural peak ΔF responses (in %ΔF/F) in the strongest responding glomeruli to different concentrations of odors for 1 week and 7 weeks old flies (n = 8 ± SEM). All GCaMP3-fluorescence responses were calculated in %ΔF/F. All p-values were calculated via Student's t-test (ns > 0.05, *p≤0.05, **p≤0.01). (C, C′) GCaMP fluorescence changes are recorded in three different responsive glomeruli (DC2, DM6 and DP1) upon stimulation with 3-octanol (12 mM). (D-D′) GCaMP fluorescence changes were measured in three responsive glomeruli (DC1, DP1 und VC2) upon stimulation with 4-methylcyclohexanol (16 mM) in 1, 4 and 6 weeks old flies. (C, D) Maximum fluorescence changes of GCaMP3 upon odor stimulation in three different glomeruli. Scale bars: 20 μm. (C′, D′) Odor-induced fluorescence change of GCaMP3 is indicated as false color images (top row) for one representative animal. Fluorescence changes over time are shown in the lower row for each different glomerulus. The pink bars represent the time window of odor presentation. n = 9; one-way ANOVA with *post hoc* Bonferroni tests. ns, not significant (p>0.05). *p<0.05. **p<0.01. ***p<0.001. (E–G) Expression of the Ca2+ sensor GCaMP3 in PNs under the control of GH146-Gal4 visualized in two focal planes in presynaptic boutons of projection neurons in calyces. Scale bars: 20 μm. (E) Representative image of *in vivo* two-photon imaging of fluorescence of GCaMP3 in PNs (GH146 > GCaMP) at their axonal extensions (boutons) in the mushroom body calyx is shown in the top image. Odor-induced fluorescence changes of GCaMP3 are indicated as false color images (bottom image) for one representative animal. (F) Maximal fluorescence changes of GCaMP3 in individual responsive boutons and (G) number of responsive boutons upon stimulation with 3-octanol (12 mM), 4-methylcyclohexanol (16 mM) or linalool (11 mM) in the two imaged focal planes. n = 9–11; one-way ANOVA with *post hoc* Bonferroni tests. ns, not significant (p>0.05). *p<0.05. **p<0.01. ***p<0.001. All traces represent mean ±SEM of ΔF/F values. Box plots indicate means, medians, interquartile ranges, and 1–99% ranges.

DOI: https://doi.org/10.7554/eLife.32018.005

The following source data and figure supplements are available for figure 2:

*Figure 2 continued on next page*

*Figure 2 continued*

**Source data 1.** Source data for *Figure 2* and *Figure 2—figure supplements 1—3*.
DOI: https://doi.org/10.7554/eLife.32018.009
**Figure supplement 1.** Olfactory receptor neuron are not affected by aging.
DOI: https://doi.org/10.7554/eLife.32018.006
**Figure supplement 2.** Olfactory receptor neurons still respond to odors in aged animals.
DOI: https://doi.org/10.7554/eLife.32018.007
**Figure supplement 3.** Older flies show strongly improved behavior to higher odor concentrations.
DOI: https://doi.org/10.7554/eLife.32018.008

LH as judged by antibody staining was unchanged (see methods; *Figure 3D*). Similarly, the expression of the GCaMP reporter (normalized to Ncad staining) used for calcium imaging, was comparable, or even slightly increased in PN axon terminals of old as compared to young flies (*Figure 3E*). Therefore, the changes observed in PN odor responses are not likely to be attributable to gross morphological changes in PN anatomy.

We next focused on expression and localization of markers of neuronal function to pinpoint a cellular defect in the PNs that could explain the observed functional decline. As a reduction in cholinergic signaling accompanies most forms of neurodegeneration and natural aging (*Doty, 2012*), and because PNs are cholinergic, we analyzed the expression of genes involved in cholinergic neurotransmission with RNA-sequencing. Our analysis revealed a significant decline in mRNA abundance of several Acetylcholine receptor (AChR) subunits (*Figure 3F*). By contrast, some previously implicated marker genes of systemic aging and lifespan did not change or increased in their expression (*Figure 3F*). Expression levels are only one factor that might impact on cellular function. The proper transport and localization of relevant proteins represents another critical point. As a proxy for synaptic integrity of the PNs, we expressed transgenic reporter constructs producing synapse-localized fluorescent marker proteins. Such a reporter construct for the localization of Acetylcholine receptors expressed in PNs (GH146 > Dα7-GFP) revealed a moderate, but significant signal reduction of this postsynaptic marker at PN postsynaptic sites in the AL indicating an aging-related change at the post-synapse (*Figure 3G,I*).

It has been proposed that axonal degeneration precedes and possibly leads to eventual neuronal loss in neurodegenerative diseases (*Kurowska et al., 2016*). Indeed, we observed that the Dα7-GFP reporter construct, which in young animals localizes not only to the post-synapse, but also, albeit to a markedly lower extend, to the axon and presynaptic terminals of PNs (in 20/20 animals analyzed) was completely absent from presynaptic terminals and axons in old flies (0/20 flies showed reporter labeling in axon, MB calyx (cx) and lateral horn (LH)) (*Figure 3G*). In mammals, AChRs are also found post- and pre-synaptically, where they modulate and enhance synaptic signaling (*MacDermott et al., 1999*).

Given the dramatic reduction of responsive presynaptic PN boutons in the MB calyx and the reduced response in the AL observed by GCaMP imaging (see *Figure 2*), we employed antibody staining against the enzyme ChAT (Choline Acetyltransferase) required for the production of Acetylcholine at synapses of cholinergic neurons such as the PNs. Quantification of these stained brains indeed revealed a significant reduction in ChAT positive puncta in the AL and at the level of the MB calyx (*Figure 3J–L*). ChAT staining in the LH, by contrast, remained relatively stable (*Figure 3L*). Beyond a change in synaptic proteins in the aged synapse, several studies in animal models including flies implicated mitochondrial dysfunction in neuronal, in particular axonal degeneration (*Court and Coleman, 2012*; *Humphrey et al., 2012*; *Valadas et al., 2015*). The current hypotheses for why axonal mitochondria are more vulnerable include the remote location of mitochondria in presynapses from the cell body, and a high metabolic activity including calcium homeostasis (*Court and Coleman, 2012*). Indeed, we observed a strong reduction in the number of puncta that were positive for a reporter construct for mitochondria (GH146 > mito-mcherry; (*Vagnoni and Bullock, 2016*) at the level of the MB calyx (*Figure 3K,L*), but not in the AL or LH area (*Figure 3J,L*). These results suggests that mitochondria are indeed depleted, lost, or not replenished selectively close to the remote synapse in the calyx, while postsynaptic sites of the PNs in the AL are seemingly maintained. Why LH synapses are not affected as compared to calyx synapses, given that they are actually further away from the cell bodies than their calycal counterparts, remains currently unknown. Nevertheless,

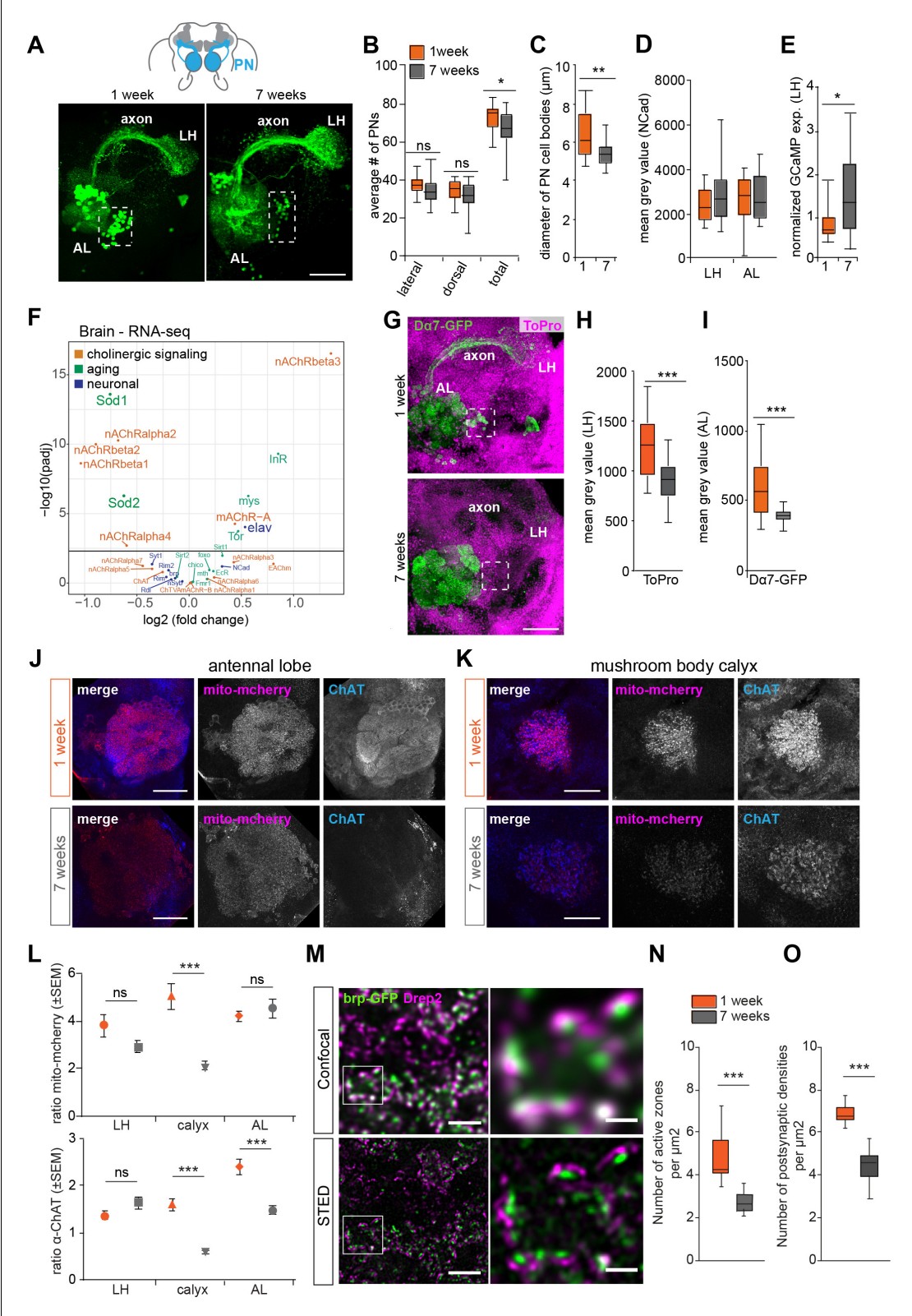

**Figure 3.** Changes in axon and synapse integrity could affect projection neuron function. (**A**) Projection neurons (PNs) of a 1 and a 7 weeks old brains are labeled with a reporter line (*GH146-Gal4;UASGCaMP3* or *GH146 >GCaMP3*) and stained with an anti-GFP antibody (green). AL, antennal lobe; cx, mushroom body calyx; LH, lateral horn; lateral cell body cluster is shown in dotted box. Scale bar: 25 µm. (**B**) Average number of PNs in the lateral, dorsal clusters and the total of both clusters. Orange boxes represent young flies (1 week), while grey boxes represent old flies (7 weeks) in all figures. *Figure 3 continued on next page*

*Figure 3 continued*

There is a mild but significant decrease in the number of reporter-labeled PNs in aged flies (Students t-test, n = 19–21) (C) Average diameter of projection neuron cell body sizes of 1 (orange) and 7 (grey) weeks old flies. The cell bodies of PNs of aged flies are significantly smaller (Students t-test, n = 19–21). (D) The box plot shows that there is no change in the expression of NCad in the defined areas for quantification (antibody staining against N-cadherin as a synaptic marker) in 1 (orange) and 7 (grey) weeks old flies AL and LH (Students t-test, n = 19–21). (E) Normalized expression levels of the GCaMP reporter protein in PNs (*GH146 >GCaMP3*) in young and old flies. The expression was normalized to Ncad antibody staining. There is no reduction of GCaMP expression in old as compared to young flies, but instead a slight but significant increase (Students t-test, n = 19–21). (F) Volcano plot of RNA-sequencing data of selected genes displaying the genes that are downregulated and upregulated in 7 weeks old brains compared to 1 week old brains, respectively. Only genes above the cutoff of –log10 (p-value adjusted (padj)) are considered significantly changed (above black line). While several AChR receptors were significantly downregulated in the brain, this was not the case in the antenna (*Figure 4—figure supplement 2C*). In addition, several aging-related genes are upregulated in older brains. Selected genes are displayed and were color-labeled by gene ontology analysis (orange: aging; green: neuronal function). (G) Reporter construct showing the localization of acetylcholine receptor (AChR) Dα7 (*GH146-Gal4;UAS-Dα7-GFP*, stained with anti-GFP antibody (green)) and ToPro nuclear marker (shown in pink) in the AL and lateral horn (LH). There is a decline at PN postsynaptic sites in the AL supporting an aging-related decline in the integrity of cholinergic synapses. For instance, the localization of Dα7 at presynaptic terminals and axons is lost in old flies (n = 20/20) in contrast to young animals (n = 0/20). See missing signal in axon and presynaptic terminals in the MB calyx and LH. Scale bar: 25 μm (H) Quantification of mean gray value (MGV) of ToPro staining of cell bodies in the area of the LH revealed a decrease in the number of cells in old as compared to young flies (n = 20). (I) A box plot shows a significant reduction in the AChR Dα7 reporter construct signal (mean grey value, MGV) of 7 weeks old flies (grey) compared to 1 week old flies (orange) at the level of the AL. Box plots show median and upper/lower quartiles. All p-values represent: ns > 0.05, *p≤0.05, **p≤0.01, ***p≤0.001. (J) Representative images of antennal lobes of 1 and 7 weeks old flies. Brains express the reporter mito-mcherry in PNs (*GH146-Gal4;UAS-mito-mcherry*; anti-RFP, red) and are stained for anti-ChAT (blue). (K) Representative images of the mushroom body calyx of 1 and 7 weeks old flies. (L) Quantification of relative expression of a mitochondria reporter (*GH146-Gal4;UAS-mito-mcherry*) and ChAT in AL, LH, and calyx. Note that mitochondria and ChAT staining are significantly reduced in the MB calyx as compared to an mito-mcherry or ChAT expression in other parts of the brain in old flies as compared to younger animals (see methods). This suggests that ChAT does not decrease equally in all brain parts, but in particular in areas such as the MB calyx. Graphs display mean relative levels ± SEM. Student's t-test: ns > 0.05, *p≤0.05, **p≤0.01, ***p≤0.001. (M) Confocal and high-resolution STED microscopy images in the calyx of flies expressing BRP-short[GFP] under control of GH146-Gal4 driver line. Green and magenta represent anti-GFP and anti-Drep2[C-Term] immunostaining, respectively. White squares in (M, left column) indicate the magnified region in (M, right column). Scale bars represent 2 μm in (M, left) and 0.5 μm in (M, right). (N) Number of active zones and (O) postsynaptic densities significantly decrease upon aging. n = 10–12; Student's t-test. ***p<0.001. Box plots indicate means, medians, interquartile ranges, and 1–99% ranges.

DOI: https://doi.org/10.7554/eLife.32018.010

The following source data is available for figure 3:

**Source data 1.** Source data for *Figure 3*.

DOI: https://doi.org/10.7554/eLife.32018.011

specific metabolic or synaptic characteristics of these different synapses and their postsynaptic partners could be part of the explanation.

In the light of an apparent reduction of ChAT and mitochondria (or at least a mitochondria reporter), we analyzed the integrity of these boutons in more detail using the high-resolution microscopy technique STED (Stimulated emission depletion; [*Kittel et al., 2006*; *Willig et al., 2006*]). To this end, the expression of a GFP-tagged version of the presynaptic protein bruchpilot was expressed in PNs (GH146 > BRP-short[GFP]; *Figure 3M*). In line with the strong reduction of odor-responsive boutons, quantitative STED analysis of this reporter revealed a strong decrease in the density of presynaptic active zones over the MB calyx (*Figure 3N*). Correspondingly, the number of postsynaptic densities as revealed by antibody staining against the postsynaptic density marker Drep2 was reduced by a similar margin (*Figure 3O*) (*Andlauer et al., 2014*).

These data indicate that a degeneration of cholinergic PNs, in particular of their axon, their presynaptic boutons and the corresponding synapses (i.e. active zones and postsynaptic densities), resembling aspects of neurodegeneration in humans, could be involved in the loss of the sense of smell.

Having identified a potential neuronal target of aging in the olfactory system, we next addressed the genetic mechanisms underpinning aging-associated olfactory decline. Several key genes and mechanisms have been identified in different model systems that contribute to systemic aging (*López-Otín et al., 2013*). To assess the mechanism of olfactory aging in our case, we used in vivo RNAi to knock-down the expression of candidate genes previously implicated in systemic aging and lifespan in the entire nervous system using a pan-neural transgenic driver line, *elav-Gal4*, or loss of function mutants. Of the ~20 genes analyzed the knock-down of only one gene, *SOD2*

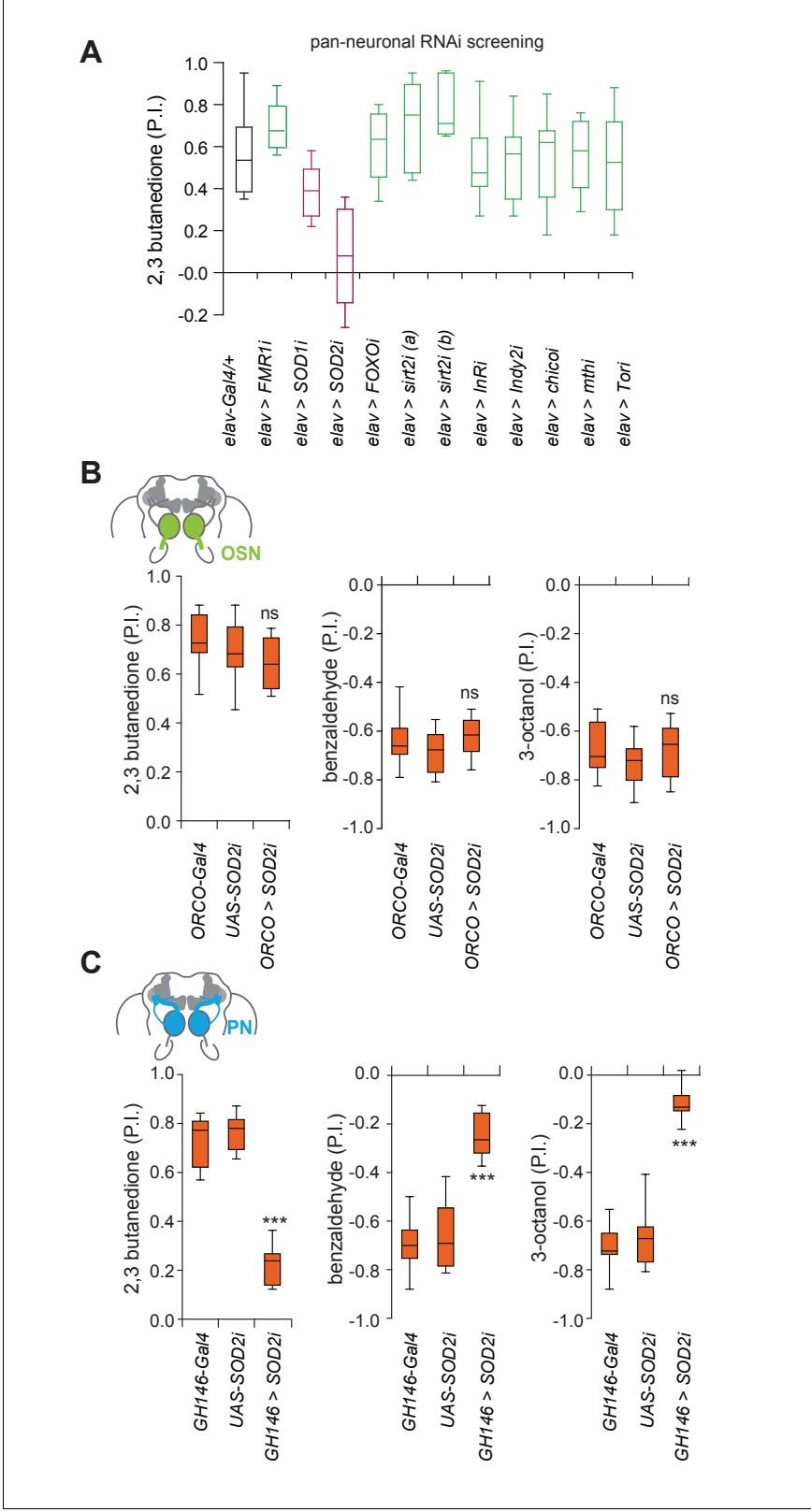

**Figure 4.** Superoxide dismutase two is selectively required in projection neurons. (**A**) RNAi knockdown behavioral screening of selected candidate genes involved in systemic aging, in the T-maze assay. RNAi Knockdown of *SOD2* (superoxide dismutase 2) pan-neuronally using the line elav-Gal4 (*elav >SOD2* i) in 3 weeks old flies significantly reduced olfactory attraction to 2,3 butanedione. In addition, two mutants (3 weeks of age) of autophagy genes did
*Figure 4 continued on next page*

*Figure 4 continued*

not show a defect in olfactory preference behavior (see *Figure 4—figure supplement 1A*) (B) RNAi knock-down of SOD2 in OSNs had no effect on the flies' olfactory preference suggesting that OSNs might be less sensitive to oxidative stress. Box plots show olfactory PIs of 1 week old flies expressing an RNAi knock-down construct for SOD2 under the control of *ORCO*, which is expressed broadly in OSNs (*ORCO-Gal4;UAS-SOD2-RNAi*) and their genetic controls to 2,3-butanedione, benzaldehyde and 3-octanol in the T-maze assay. (C) RNAi knock-down of SOD2 in PNs using the GH146-Gal4 (*GH146-Gal4;UAS-SOD2-RNAi*), results in strongly diminished olfactory preference of flies to 2,3-butanedione, benzaldehyde and 3-octanol. Box plots show median and upper/lower quartiles (n = 8, 60 flies/trial, 30 ♀ and 30 ♂). All p-values were calculated via one-way ANOVA with the Bonferroni multiple comparison posthoc test (ns >0.05, *p≤0.05, **p≤0.01, ***p≤0.001).

DOI: https://doi.org/10.7554/eLife.32018.012
The following source data and figure supplements are available for figure 4:

**Source data 1.** Source data for *Figure 4* and *Figure 4—figure supplements 1* and *3*.
DOI: https://doi.org/10.7554/eLife.32018.016
**Figure supplement 1.** Autophagy mutants show normal olfactory behavior.
DOI: https://doi.org/10.7554/eLife.32018.013
**Figure supplement 2.** SOD1 and SOD2 are downregulated in aged brains.
DOI: https://doi.org/10.7554/eLife.32018.014
**Figure supplement 3.** Lifespan of experimental and control groups.
DOI: https://doi.org/10.7554/eLife.32018.015

(mitochondrial form of superoxide dismutase) significantly affected olfactory behavior in young animals (*Figure 4A*; *Figure 4—figure supplement 1A,B*). By contrast, knock-down of the cytoplasmic form, *SOD1*, only resulted in a smaller but non-significant decrease in olfactory behavior (*Figure 4A*). Sirt2 RNAi knock-down led to a mild, but not significant increase in odor attraction, possibly reflecting the fact that Sirt2 is upregulated in older brains (*Figure 4A*). More generally, it will be interesting to test overexpression of genes upregulated in aged brain for their role in neurodegeneration in the future.

SODs protect against reactive oxygen species (ROS) and SOD2 appears to be required for normal lifespan (*Kirby et al., 2002*; *Oka et al., 2015*). Our RNA-seq data revealed that while *SOD1* and *SOD2* expression remained unchanged in the antennae of old flies, both genes were expressed at lower levels in older brains (*Figure 4—figure supplement 2A–C*).

In addition to SOD1 and SOD2, other genes were significantly up- and downregulated in older brains. Among the most significant up-regulated GO terms were proteolysis and defense/immunity genes, while energy metabolism and oxidation process related genes ranked high in the list of downregulated genes. For instance, genes belonging to the GO term 'oxidation-reduction process' were found to be significantly downregulated (FDR < 0.01), which corresponds to 128% more genes than expected by chance (p=2.3e-14, Fisher Test). Similarly, 119 genes belonging to proteolysis GO term were found to be significantly upregulated (FDR < 0.01), which corresponds to 59% more genes than expected by chance (p=1.3e-8, Fisher Test) (*Supplementary file 1* and *2*). These results were also well in line with our finding of a reduction of mitochondria in PN synapses (see *Figure 3*).

We next tested whether any particular neuron type in the olfactory system was most vulnerable to oxidative stress, or whether this was a systemic effect. Notably, knock-down of *SOD2* in OSNs (ORCO-Gal4) had no effect on the flies' olfactory attraction or aversion compared to genetic controls (*Figure 4B*). By contrast, knock-down of *SOD2* in PNs resulted in a significantly reduced odor attraction and odor aversion similar to the knock-down of *SOD2* in all mature neurons (*Figure 4C* and *Figure 4—figure supplement 2D*). While ubiquitous or pan-neural knock-down of *SOD2* with the same construct as used here is lethal (flies die after ~3 days; (*Kirby et al., 2002*) and our own observations), knock-down of *SOD2* in PNs did not reduce the flies' life span significantly as compared to the genetic controls, which were also used in the behavioral experiments (*Figure 4—figure supplement 3A*). It is important to note, however, that given the well-known effects of genetic background on lifespan, we limit our conclusion to the correlation of the lifespan of each individual test or control group to their respective olfactory behavioral performance. In other words, an extremely short- or long-lived animal might show unspecific reasons for behavioral deficits or improvements. Based on

this, a reduced lifespan is unlikely to explain the strong behavioral phenotype, suggesting a high vulnerability of PNs and their central role in olfactory decline.

Does lack of SOD2 in PNs result in changes at the level of neuronal function and morphology similar to what we observed for natural aging? To answer this, we carried out some of the same analysis as done for aged flies (see *Figures 2* and *3*). First, we counted the number of PNs in 1 week old flies where SOD2 was knocked-down under the control of *GH146-Gal4* (GH146 > SOD2 i). In contrast to the aged flies, the number of GCaMP reporter expressing PNs (GH146 > GCaMP; with or without SOD2i) was not significantly different between the experimental group and controls (*Figure 5A,B*). However, the size of the PN cell bodies was significantly smaller upon knock-down of SOD2 as compared to controls (*Figure 5C*), similar to the decline of cell body size in old flies (see *Figure 3*). Quantification of the expression of GCaMP stained with an anti-GFP antibody in the LH and calyx revealed a small reduction, which was significant in the LH but not in the calyx, suggesting that the relative expression level of GCaMP upon normalization to staining outside the LH and calyx, respectively, remained at a similar level as compared to controls (*Figure 5D*). By contrast, antibody staining against ChAT showed that the expression of this enzyme was significantly reduced in PN boutons in the MB calyx (*Figure 5E*) indicating a reduction of functional cholinergic synapses – again in line with the results in aged brains.

Next, we tested what the consequences of SOD2 lack meant for PN function by using two photon in vivo calcium imaging of the PN boutons in the MB calyx (*Figure 5F–H*). While SOD2 knock-down did not affect the responses of individual PN boutons to odor stimulation, the number of responsive boutons was significantly reduced upon SOD2 knock-down in PNs (*Figure 5G,H*).

Taken together, reduction of SOD2 expression exclusively in PNs leads to behavioral, functional and anatomical phenotypes that resemble several of the phenotypes observed in naturally aged brains.

Systemic SOD2 overexpression can ameliorate memory deficits in a transgenic Alzheimer's disease mouse model (*Massaad et al., 2009*). However, it is not known whether augmenting SOD2 exclusively in one neuron type, the so to speak potential seed or Achilles heel of degeneration, prevents aging-associated decline of an entire circuit and its ability to control and drive behavior. We found that overexpression of SOD2 exclusively in PNs fully rescued behavioral decline of 7 weeks old flies, which behaved just like their 1 week old genetic counterparts (*Figure 6A* and *Figure 6— figure supplement 1A*). Importantly, SOD2 overexpression in PNs did not significantly extend or shorten the average lifespan of this group of flies as compared to the used genetic controls, which were also used for behavioral analysis (*Figure 4—figure supplement 3B*). Although these results do not allow strong conclusions regarding the effect on lifespan, they do, nevertheless, indicate a more specific role of SOD2 in PNs, and that the behavioral improvement is unlikely to be a result of an improved overall fitness of these animals. Furthermore, SOD2 expression under the control of ORCO-Gal4 in OSNs did not rescue the behavior of old flies (*Figure 6B* and *Figure 6—figure supplement 1B*). We conclude that cholinergic PNs represent key targets and possibly a 'seed neuron' population in aging-associated decline of the sense of smell. These results suggest that aging-related degeneration and behavioral decline could be significantly delayed by preventing oxidative damage in only one or few neuron types (see also [*Seeley, 2017*]).

The overexpression of SOD2 does currently not provide a feasible way to protect the nervous system of humans during aging or during the onset or course of a neurodegenerative disease. A popular idea is that it might be possible to boost an organism's ability to fight ROS by consuming a diet high in antioxidants (*Vaiserman and Marotta, 2016*). We found that a diet high in Resveratrol, a well-studied antioxidant shown to increase the expression of SOD2 in neurons (*Fukui et al., 2010*) and with potential benefits against AD (*Granzotto and Zatta, 2014*), protected from olfactory decline. The relatively small effect on the flies' lifespan (*Wood et al., 2004*) (*Figure 4—figure supplement 3C*) is unlikely to explain the observed behavioral improvement (*Figure 6C* and *Figure 6— figure supplement 1C*). Notably, a 1 week Resveratrol treatment of younger flies did not affect olfactory behavior as compared to solvent fed flies (*Figure 6—figure supplement 1D*) indicating that Resveratrol might indeed counteract oxidative stress that builds up during aging. Thus, protection from oxidative stress, plausibly at least in part through protection of PNs, might help to maintain the function of the olfactory system in an aged individual.

Apart from certain diets and nutrients, the gut microbiome has been implicated in progression of Parkinson's disease and aging (*Kong et al., 2016*; *Scheperjans et al., 2015*). Recent studies show

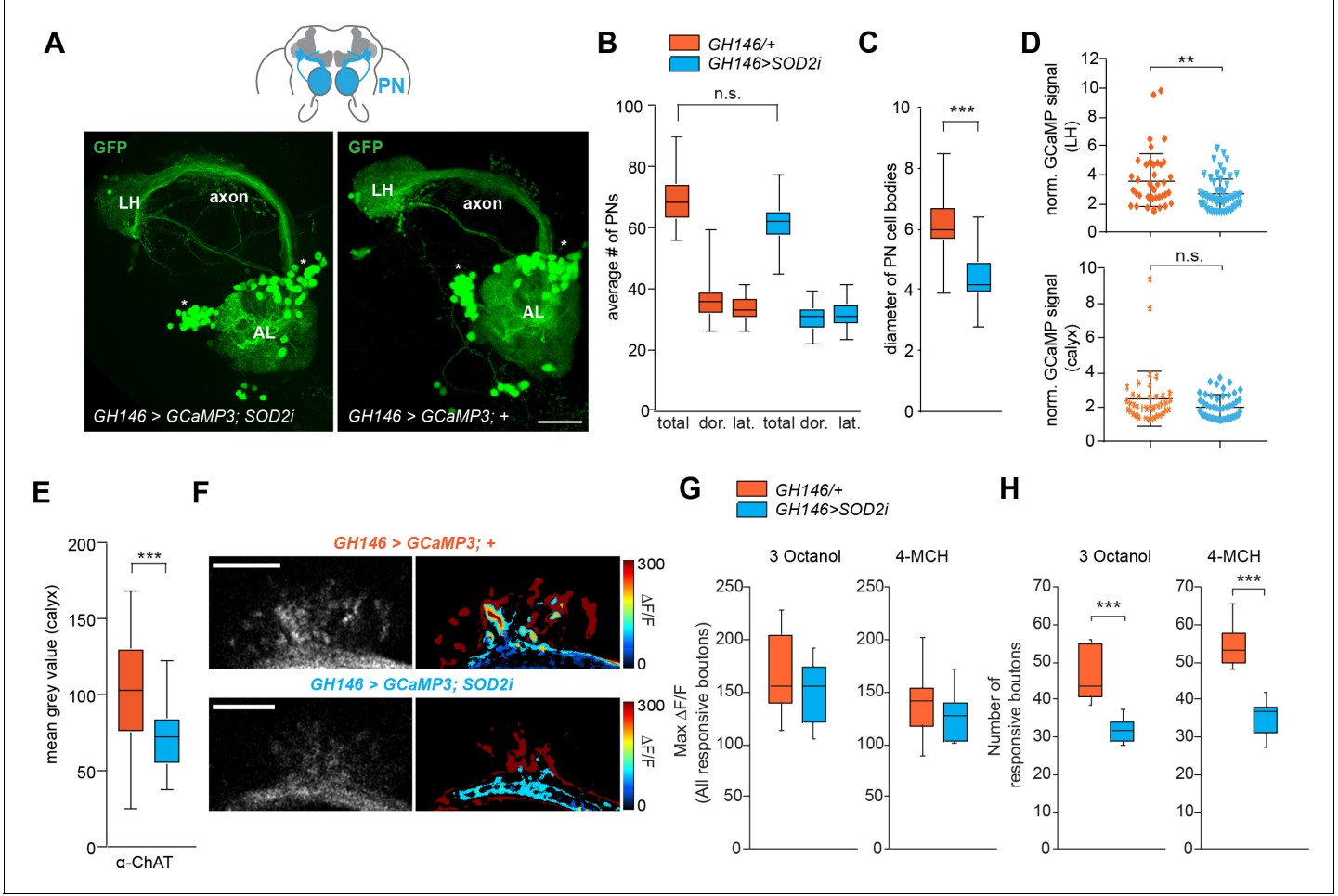

**Figure 5.** SOD2 deprived PNs resemble neurons in aged brains. (**A**) SOD2 RNAi (*GH146-Gal4;UAS-SOD2i*) expressing and SOD2 RNAi negative controls labeled with the reporter line (*GH146-Gal4;UASGCaMP3* or *GH146 >GCaMP*) and stained with an anti-GFP antibody (green). AL, antennal lobe; cx, mushroom body calyx; LH, lateral horn; lateral cell body cluster is shown in dotted box. Scale bar: 25 μm. (**B**) Average number of PNs in the lateral, dorsal clusters and the total of both clusters. Orange boxes represent 1 week control flies, while blue boxes represent flies carrying GH146 >SOD2 i in all figures. There is no significant decrease in the number of reporter-labeled PNs upon SOD2 knock-down (Students t-test, n = 19–21) (**C**) Average diameter of projection neuron cell body sizes of controls (orange) and SOD2 knock-down flies (blue). The cell bodies of PNs are significantly smaller when SOD2 is reduced exclusively in PNs (Student's t-test, n = 19–21). (**D**) Scatter plots showing normalized GCaMP signal stained with α-GFP antibody (MGV). The intensity of staining within the LH (upper panel) and within the calyx (bottom panel) was normalized to the background signal in a non-GFP positive brain area of the same brain (Student's t-test, n = 28). (**E**) Mean gray value (MGV) of anti-ChAT antibody staining in the MB calyx. Note that knocking-down SOD2 in PNs significantly reduced the ChAT signal (Student's t-test, n = 28). (**F**) Representative image of in vivo two-photon imaging of fluorescence of GCaMP3 in PNs (GH146 >GCaMP) at their axonal extensions (boutons) in the mushroom body calyx for test (*GH146 >GCaMP;SOD2i*) and control flies (*GH146 >GCaMP;+*). Odor-induced fluorescence change of GCaMP3 are indicated as false color images (right column) for one representative animal of each genotype. Scale bars: 20 μm. (**G**) maximal fluorescence changes of GCaMP3 in individual responsive boutons and (**H**) number of responsive boutons upon stimulation with 3-octanol (12 mM) or 4-methylcyclohexanol (16 mM) in the two imaged focal planes. n = 10; Student's t-test. (ns >0.05, *p≤0.05, **p≤0.01, ***p≤0.001). Box plots indicate means, medians, interquartile ranges, and 1–99% ranges.

DOI: https://doi.org/10.7554/eLife.32018.017

The following source data is available for figure 5:

**Source data 1.** Source data for *Figure 5*.

DOI: https://doi.org/10.7554/eLife.32018.018

that beneficial effects of microbiota are conserved between *Drosophila* and mouse (*Schwarzer et al., 2016*). For instance, the effects of malnutrition can be partially overcome by inoculating flies with a specific strain of *Lactobacillus plantarum* (*L.p.WJL*) or *Acetobacter pomorum* (*A. p.*) (*Schwarzer et al., 2016*). Importantly, another strain of *L. plantarum* (*L.p.NI202877*) did not

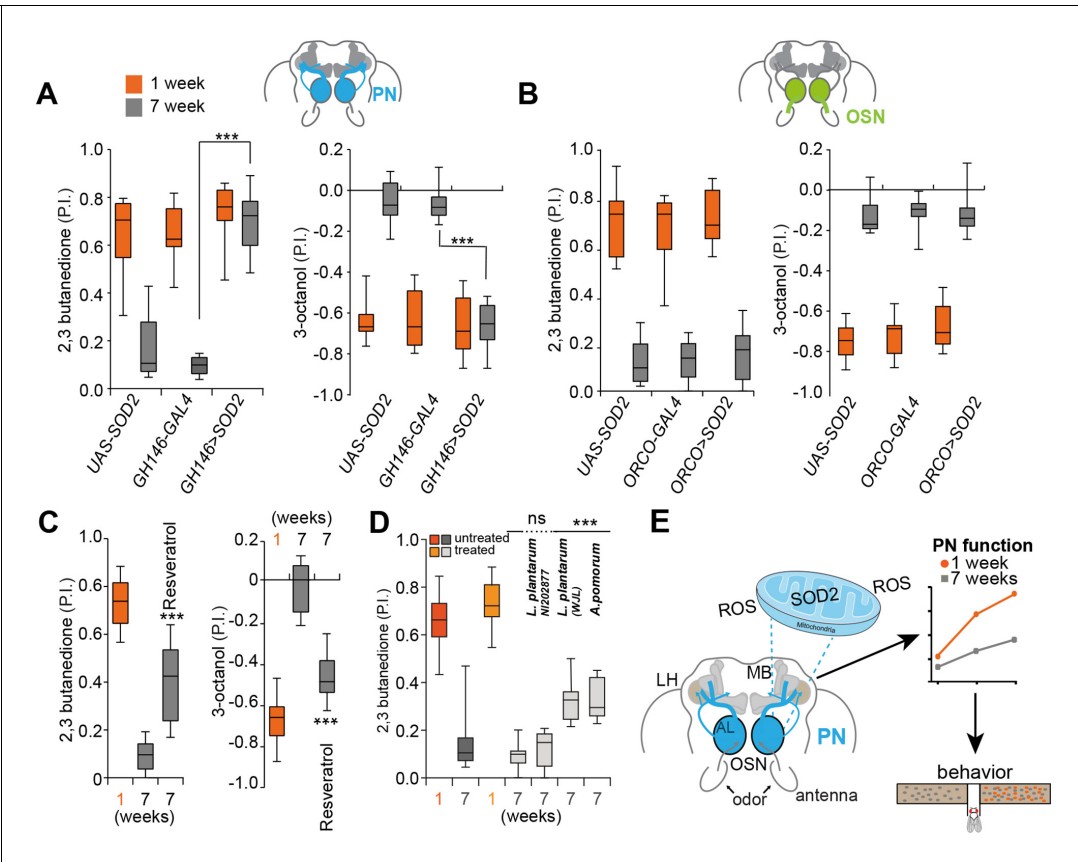

**Figure 6.** Expression of SOD2 in single neuron subtype fully rescues olfactory degeneration. (**A**) Box plots show PIs of 1 week old (orange boxes) and 7 weeks old (grey boxes) transgenic flies overexpressing SOD2 exclusively in projection neurons (PNs) (*GH146-Gal4;UAS-SOD2*) and their genetic controls in response to attractive (2,3-butanedione) and aversive (3-octanol) odors. Note that expression of SOD2 exclusively in PNs fully rescues olfactory performance in 7 weeks old flies indicating that sensitivity to oxidative stress of PNs represents a key player in the aging-associated decline of the olfactory system. (**B**) Box plots show PIs of 1 week old (orange) and 7 weeks old (grey) transgenic flies overexpressing SOD2 under the control of *ORCO-Gal4* in OSNs (*ORCO-Gal4;UAS-SOD2*) and their genetic controls in response to attractive (2,3-butanedione) and aversive (3-octanol) odors. Importantly, overexpression of SOD2 under the control of *ORCO-Gal4* in OSNs had no effect on the behavior of old flies. (**C**) Box plots show preference of 1 week old and 7 weeks old flies raised on standard fly food (first two boxes) and 7 weeks old flies raised on standard fly food mixed with Resveratrol (third grey box) in response to attractive (2,3-butanedione) and aversive (3-octanol) odors. All p-values were calculated via t-test (ns >0.05, *p≤0.05, **p≤0.01, ***p≤0.001). In all figures, asterisks above a single bar refer to p-values of comparison to the control (7 weeks old flies in second bar). (**D**) Box plots show untreated and treated (flies were inoculated with the indicated bacterial strain after being pretreated to become germfree) 1 week old (light (treated) and dark orange (not germfree, standard conditions as in all other experiments before) and 7 weeks old (dark and light grey) flies. Treated flies were inoculated with *Lactobacillus plantarum NI202877, L. plantarum WJL, Acetobacter pomorum (A.p.)*, while control flies were just raised on standard fly food. P.I.s of flies to the attractive odor 2,3-butanedione is shown. All p-values were calculated via two-way ANOVA with the Bonferroni multiple comparison posthoc test (ns >0.05, *p≤0.05, **p≤0.01, ***p≤0.001). Asterisks above a single box refer to p-values of comparison to the control (7 weeks old treated control). Box plot show median and upper/lower quartiles (n = 8, 60 flies/trial 30 ♀ and 30 ♂). (**E**) Summary and model of presented results. SOD2, the mitochondrial form of SOD, protects projection neurons (PN) from oxidative stress. Their vulnerability to oxidative stress and reactive oxygen species (ROS) appears to be the weak point of the olfactory system of Drosophila. The decline in function of PNs ultimately results in strongly reduced sensitivity to odors and accordingly diminishes behavioural responses.

DOI: https://doi.org/10.7554/eLife.32018.019

The following source data and figure supplement are available for figure 6:

**Source data 1.** Source data for *Figure 6* and *Figure 6—figure supplement 1*.
DOI: https://doi.org/10.7554/eLife.32018.021

**Figure supplement 1.** SOD2 and antioxidants slow down aging-associated olfactory decline.
DOI: https://doi.org/10.7554/eLife.32018.020

produce the same effect suggesting a specific mechanism and not just the presence of any bacteria in the gut. Oral administration of *L. plantarum*, which colonizes the mammalian gut, led to a significant increase of SOD levels in serum and liver in a mouse aging model (*Tang et al., 2016*). Moreover, *L.p.WJL* modulates TOR and insulin signaling in *Drosophila*, both of which are regulators of lifespan (*Storelli et al., 2011*). We tested the effect of these microbiota on olfactory aging. *L.p.WJL* and *A.p.*, but not *L.p.NI202877*, improved the old flies' performance in olfactory preference assays significantly (*Figure 6D* and *Figure 6—figure supplement 1E*). We conclude that certain microbiota have a positive effect and slow-down aging-associated olfactory neurodegeneration.

Based on the data presented, we propose that due to their insufficient resistance to oxidative stress, which is facilitated by an aging-associated decrease in SOD levels, functional degeneration starting at the axon of specific cholinergic neurons is responsible for the decline of the olfactory circuit and the sense of smell (*Figure 6E*). A central role of cholinergic neurons in neurodegeneration seems conserved. In *C. elegans*, a decline in cholinergic signaling likely triggers an aging-associated decline in the sense of smell (*Leinwand et al., 2015*). In humans, cholinesterase inhibitors, which augment levels of acetylcholine in the brain, represent the main class of Alzheimer drugs (*Canter et al., 2016*). Using the *Drosophila* model, we showed that a specific type of cholinergic neurons plays a key role in the loss of the sense of smell. Our data therefore provides experimental evidence that declines in nervous system function are not due to a universal degeneration, but rather that specific neuronal subsets are primarily or even solely responsible and might trigger further degeneration throughout a neuronal network.

Why are certain neurons more sensitive than others? Similar to flies, axonal degeneration and decline of cholinergic neurotransmission play important roles in neurodegeneration in humans. Indeed, neurons with long axonal projections, broad input, high sensitivity and high action potential frequency, such as PNs (*Wilson, 2013*), might be particularly vulnerable to aging. The olfactory system of the fly could help to pinpoint the 'Achilles heel' of a neuron and aid the development of more targeted treatments by combining high-throughput genetic screening with drug or microbiota treatments.

## Materials and methods

### Key resources: Fly rearing and lines

*Drosophila melanogaster* stocks were raised on conventional cornmeal-agar medium at 25°C temperature and 60% humidity and a 12 hr light:12 hr dark cycle. The following fly lines were used to obtain experimental groups of flies for the different experiments:

1. *Canton S*
2. *w^1118^*
3. *Or42b-Gal4 (Bloomington stock (BL) # 9971)*
4. *UAS-mCD8GFP (gift from L. Zipursky)*
5. *GH146-Gal4 (gift from L. Zipursky)*
6. *elav-Gal4;UAS-Dcr-2 (BL# 25750)*
7. *RNAi: UAS-SOD2i (Bloomington stock (BL) # 24489; [Kirby et al., 2002]), UAS-SOD1i (BL# 24493), UAS-TORi (BL# 33627), UAS-InRi (VDRC ID 992), UAS-Sirt2i(a, BL#31613), UAS-Sirt2i(b, BL# 36868), UAS-Indyi (VDRC ID 9981), UAS-chicoi (BL# 28329), UAS-mthi (BL# 27495), UAS-FOXOi (VDRC ID 107786), UAS-FMR1i (BL# 27484)*
8. *UAS-SOD2 (BL# 24494)*
9. *nsyb-Gal4 (gift from L. Zipursky)*
10. *UAS-GCaMP3 (gift from S. Sachse)*
11. *ORCO-Gal4 (BL# 23292)*
12. *Orco^1^ (BL# 23129)*
13. *UAS-Dα7-GFP (gift from G. Tavosanis)*
14. *UAS-BRP-short^GFP^ (Christiansen et al., 2011); Kremer et al., 2010)*
15. *y[1], Atg8a[d4] (from [Pircs et al., 2012])*
16. *w[1118]; Atg7[d77] (from [Juhász et al., 2007])*

The lines were obtained from Bloomington (http://flystocks.bio.indiana.edu/) or the Vienna Drosophila Resource Center (VDRC) stock center (http://stockcenter.vdrc.at/control/main) unless indicated otherwise.

### Further key resources

1. Resveratrol (Sigma-Aldrich, R5010)
2. *Lactobacillus plantarum* or *Acetobacter pomorum* (all strains were a gift of Francois Leulier)
3. Antibodies and additional reagents used for molecular biology are described in the respective methods section.

## Aging and recording of lifespan

100 flies (50 males and 50 females) were flipped onto fresh food every second day for up to 10 weeks. For lifespan recordings, flies were counted every other day or once a week. Please note that conclusions regarding lifespan might be confounded by genetic background etc. as flies have not been backcrossed for 10 generations or more. However, the lifespan experiments were carried out solely to be able to correlate the lifespan of each group of flies to its respective behavioral, physiological, or anatomical phenotype.

## Olfactory T-Maze assay

1–10 weeks old flies collected on the same day were used for all experiments and tested on the same day side by side. For experiments with RNAi, experimental flies and genetic controls were raised at 30°C to enhance the effect of the RNAi. Flies were tested in groups of ~60 (30 females and 30 males) in a T-maze and were allowed 1 min to make a decision to go into either arm. Experimentation was carried out within climate-controlled boxes at 25°C and 60% rH in the dark. 50 µl of fresh odor solution (all odors were purchased at Sigma-Aldrich) at different concentrations diluted in distilled water or paraffin oil applied on Whatman chromatography paper was provided in the odor tube, except for 1% $CO_2$, which was diluted using a custom-build setup with mass flow controller from pure $CO_2$ and bottled air. Control tubes were filled with 50 µl odorant solvent or compressed air in the case of $CO_2$. Unless otherwise indicated, 1 mM of odor dilution was used. After experimentation, the number of flies in each tube was counted. An olfactory preference index (P.I.) was calculated by subtracting the number of flies on the test odor site from the number of flies on the control site and normalizing by the total number of flies. Statistical analysis was performed using ANOVA and the Bonferroni multiple comparisons posthoc test using Prism GraphPad 6.

## Visual T-maze assay

Experiments were essentially carried out as described above, but with a visual instead of an olfactory stimulus. A modified transparent T-maze apparatus was used to allow stimulation with light. The two arms of the T-maze were illuminated with blue LED emitting lights (465–470 nm) on one side or red LED lights (625–630 nm) on the other side. by subtracting the number of flies on the test odor site from the number of flies on the control site and normalizing by the total number of flies. Statistical analysis was performed using ANOVA and the Bonferroni multiple comparisons posthoc test using Prism GraphPad 6.

In some experiments flies were videotracked by using an infrared camera on the top of the climate box and analyzed with ctrax software.

## Treatment with *Lactobacillus plantarum* or *Acetobacter pomorum*

200 fly embryos (12 hr AEL) were collected in a petri dish, washed with 10% bleach once, in 70% ethanol twice, in PBS once, and in sterile, distilled water twice. These embryos are considered germfree and are referred to as *treated*. Embryos were then transferred into new food bottles. An overnight *L. plantarum* culture was concentrated to OD:200 and 300 µl were added into the fly food every third day.

## Treatment with resveratrol

For Resveratrol-supplemented diets, Resveratrol was dissolved in 100% ethanol and added to fly food to a final concentration of 100 µM. All diets contained 1.5% agar and equal amounts of ethanol.

## Anatomy

Adult fly brains were dissected, fixed, and stained as described previously (Hartl et al., 2011). Briefly, brains were dissected in cold PBS, fixed with paraformaldehyde (2%, overnight at 4°C or for 2 hr at RT), washed in PBS, 0.1% Triton X-100, 10% donkey serum and stained overnight at 4° C or for 2 hr at RT with the primary and after washes in PBS, 0.1% Triton X-100 with the secondary antibody using the same conditions. All microscopic observations were made at an Olympus FV-1000 or at a Leica SP8 confocal microscope. Images were processed using ImageJ and Photoshop. The following antibodies were used: chicken anti-GFP and anti-RFP (molecular probes, 1:100), rat anti-N-cadherin (anti-N-cad DN-Ex #8, Developmental Studies Hybridoma Bank, 1:100), and mouse anti-ChAT (Yasuyama et al., 1995). Secondary antibodies used were: anti-chicken Alexa 488 (molecular probes, 1:250), a-mouse Alexa 633 (molecular probes, 1:250) and anti-rabbit Alexa 549 (molecular probes, 1:250), respectively.

For image quantification, all brains were processed at the same time using the same conditions. Images were taken at the exact same settings. All analysis was done blind to the genotype or age of the flies. For the quantification of cell bodies, neurons were counted section-by-section either directly at the confocal or using ImageJ/FIJI software. For antibody staining or reporter construct expression at the level of the lateral horn (LH), images were Z-projected into a single image. Regions of interest (ROI) were drawn around the LH in each image and quantification was carried out using FIJI ImageJ software. For quantification of stainings in the antennal lobe (AL) and calyx, three sections at similar levels of the structures were chosen. ROIs were drawn around the AL or calyx in each section and combined to quantify each individual brain. For each brain, only one LH, calyx and AL were chosen randomly for quantifications. For quantification of staining intensity, different strategies were used depending on the staining. For GCaMP staining, all stainings were normalized to Ncad antibody staining. For D$\alpha$7-GFP and ToPro staining, staining intensity was measured as mean grey value (MGV) and normalized to the size of the area that was measured. For quantification of ChAT and mito-RFP staining in Figure 3, ratios of stainings were calculated by dividing the MGV in the area of interest by another brain area of equal size next to it, that is, in a region of the brain just below the calyx that did not express the protein or the reporter or that did not belong to the region of interest. With the aim to calculate how brain areas were affected by aging relative to each other. A decrease in the ratio showed that the MGV in for instance the calyx was more strongly decreased as compared to another brain area. For Figure 5, ChAT staining was quantified as MGV. Measurements of cell body size diameter were carried out in ImageJ/FIJI. 3 cell bodies were measured per section in several sections of the brain or antenna. The averages of the measured diameters were used for statistical analysis.

All statistical analysis was carried out with GraphPad Prism software. The exact statistics used in each experiment are indicated in the respective figure legends. Of note, different absolute values are the result of the use of two different confocal microscopes; the newer Leica SP8 was significantly more sensitive as compared to the older Olympus FV1000. As the result, individual settings for each microscope were used and absolute number can only be compared within each individual graph.

## Immunostaining for STED microscopy

Brains were dissected in Ringer's solution (pH 7.3, 290–310 mOsm) containing 5 mM HEPES-NaOH, 130 mM NaCl, 5 mM KCl, 2 mM MgCl2, 2 mM CaCl2, and 36 mM sucrose, fixed in 4% PFA for 2 hr at 4°C and washed three times for 10 min each in PBS containing 0.6% Triton X-100 (PBT) at room temperature. Samples were incubated for 2 hr in PBT containing 2% BSA and 5% normal goat serum. Subsequently, the samples were incubated in the primary antibody diluted in block solution at 4°C overnight. For staining BRP-short[GFP], FluoTag-X4 anti-GFP, Abberior Star 635P-conjugated (1:100, NanoTag Biotechnologies, N0304-Ab635P) and for staining postsynaptic densities of Kenyon cells, rabbit anti-Drep- 2[C-Term] (1: 500, Andlauer et al., 2014) were used. Samples were washed three times for at least 30 min each in PBT containing 2% BSA (PAT) at room temperature, subsequently incubated with secondary antibody diluted in PAT overnight at 4°C. As secondary antibody AlexaFluor-594-coupled goat anti-rabbit (1: 200, Thermo Fisher Scientific, A-11012) was used. Brains were washed at least six times for 30 min each in PBT and embedded in Prolong Gold Antifade (Invitrogen). Samples were stored for 24 hr at room temperature followed by 48 hr at 4°C, before STED microscopy.

## Two-Color STED imaging

STED imaging with time-gated detection was performed on a Leica SP8 TCS STED microscope (Leica Microsystems) equipped with a pulsed white light excitation laser (NKT Photonics). Dual-channel STED imaging was performed by sequentially exciting Abberior Star 635P and Alexa 594 at 646 nm and 598 nm, respectively. Both dyes were depleted with a 775 nm STED laser. Three optical sections at a distance of 250 nm were acquired with an HC PL APO CS2 100×/1.40 N.A. oil objective (Leica Microsystems), a scanning format of 1024 × 1024 pixel, eight bit sampling and six fold zoom, yielding a pixel dimension of 18.9 × 18.9 nm. Time-gated detection was set from 0.3 to 6 ns for all dyes. To minimize thermal drift, the microscope was housed in a heatable incubation chamber (LIS Life Imaging Services).

For quantifications, raw data obtained from STED imaging were de-convoluted using Huygens Professional (Scientific Volume Imaging, Netherlands) and analyzed by a custom-written ImageJ macro. Briefly, z-stacks of de-convoluted images were projected and each channel was segmented using the auto local threshold method 'Phansalkar' followed by watershed separation of touching segmented particles. The number, area fraction, and feret diameter from all segmented particles with sizes from 5 to 200 pixels were measured in >4 separate images per calyx region.

## In vivo calcium imaging

For calcium imaging experiments with an epifluorescence microscope, GCaMP3 was expressed under the control of GH146-Gal4 (flies were heterozygous for both transgenes). Female flies were prepared in a modified setup according to a method previously reported (*Hussain et al., 2016b2016b*). In vivo preparations were imaged using a Leica DM6000FS fluorescent microscope equipped with a 40x water immersion objective and a Leica DFC360 FX fluorescent camera. All images were acquired with the Leica LAS AF E6000 image acquisition suit. Images were acquired for 20 s at a rate of 20 frames per second with 4 × 4 binning mode. To calculate the normalized change in the relative fluorescence intensity, we used the following formula: $\Delta F/F = 100(F_n-F_0)/F_0$, where $F_n$ is the nth frame after stimulation and $F_0$ is the averaged basal fluorescence of 15 frames before stimulation. The peak fluorescence intensity change is calculated as the mean of normalized trace over a 2 s time window during the stimulation period. During all measurements, the exposure time was kept constant at 20 ms. For all experiments with odor stimulation, the stimulus was applied 5 s after the start of each measurement. A continuous and humidified airstream (2000 ml/min) was delivered to the fly throughout the experiment via an 8 mm diameter glass tube positioned 10 mm away from the preparation. A custom-made odor delivery system (Smartec, Martinsried), consisting of mass flow controllers (MFC) and solenoid valves, was used for delivering a continuous airstream and stimuli in all experiments. In all experiments, stimuli were delivered for 500 ms, and during stimulations the continuous flow was maintained at 2000 ml/min. For odorant stimulations, 1 ml of a precise concentration was filled in the odor delivery cup and the collected airspace odor was injected into the main airstream to give 0 mM, 0.1 mM, 1 mM, and 10 mM final concentrations for 500 ms without changing airstream strength. To measure the fluorescent intensity change, the region of interest was delineated by hand and the resulting time trace was used for further analysis. To calculate the normalized change in the relative fluorescence intensity, we used the following formula: $\Delta F/F = 100(F_nF_0)/F_0$, where $F_n$ is the nth frame after stimulation and $F_0$ is the averaged basal fluorescence of 15 frames before stimulation. The peak fluorescence intensity change is calculated as the mean of normalized trace over a 2 s time window during the stimulation period. The pseudo-colored images were generated in MATLAB using a custom written program. All analysis and statistical tests were done using Excel and GraphPad6 Prism software, respectively.

For live imaging using two-photon microscopy, flies were homozygous for both GH146-Gal4 and the UAS-GCaMP3 construct. Female transgenic flies (5–6 day, 4 week and 6 weeks old) were used for imaging experiments. Imaging was performed using two-photon microscope (Leica) equipped with a 20x water-immersion objective (NA = 1, Leica). GCaMP3 was excited at 920 nm. A custom-built device was used as odor delivery system to supply odors with a constant flow rate of 1 ml/s to the fly's antennae for 2 s. Onset and duration of the odor stimulus were controlled using a custom-written LABVIEW program. Images were recorded at 5 Hz. Image processing and analysis was performed using Fiji software. For correcting the potential slight movements in x-y direction, recorded images were aligned using TurboReg plugin (*Thévenaz et al., 1998*). Afterwards, regions of interest

(ROIs) were manually defined. In the antennal lobe, individual glomeruli and in the calyx, individual presynaptic boutons were selected as ROIs. For signal quantification, the average pixel intensity of five frames before stimulus onset was determined as F. ΔF is the difference between fluorescence and F, and resulting values were divided by F and displayed as percent.

## In vivo electrophysiology

Single sensillum recordings (SSR) are extracellular recordings performed from antennal olfactory sensilla as previously described (*Hartl et al., 2011*). A single fly was wedged into narrow end of a truncated 200 µl pipette tip and placed on a slide under the objective. The fly head was exposed and stabilized on top of a glass coverslip. The antennae were hold by the tip of a glass capillary. An odor delivery pipette blew continuous air-streams to the antenna providing odor stimulations of different concentrations (SYNTECH, the Netherlands). A glass reference electrode filled with ringer (0.01 mM KCl) was inserted into the fly eye gently by a micromanipulator (Sutter instruments). And a glass recording electrode filled with ringer (0.01 mM KCl) was pushed against a sensillum until it pierced the cuticular wall of the sensillum. The recording of action potentials (APs) was started after an observation of spontaneous responses of olfactory neurons. The AC signals (10–2800 Hz) of the responses were amplified 500x (Multiclamp 700B, United States) and were analyzed with Clampex10.3 software (Digidata 1440A, the United States). The signals to a particular stimulus were recorded 5 s before giving the odor stimulation. The responses of neurons were calculated by counting the number of APs for 0.5 s during the response minus the number of APs for 0.5 s before the response (spikes/s). Statistical analysis was performed by one-way ANOVA using GraphPad Prism software.

## RNA-sequencing

RNA was extracted and sequencing was carried out using standard methods and as previously described (*Hussain et al., 2016a*). RNA-sequencing was performed using the Illumina HiSeq 2000/2500 sequencer suite.

## Statistical methods for RNA-seq and gene ontology analysis

Gene expression data was normalized by size factors and tested for differential expression using DESeq2 package (v. 1.16.1) (*Love et al., 2014*) in R (v. 3.4.0). Significant up and down regulated genes were classified according to their $\log_2$ fold change and adjusted p-value<0.01. Then, up and down regulated genes were analyzed separately to obtain enriched gene ontology terms using topGO package (v. 2.28.0) in R and the *D. melanogaster* gene ontology annotation database gene_association.fb (http://www.flybase.org).

## Acknowledgements

We would like to thank members of the Grunwald Kadow lab for critically reading the manuscript. We thank Francois Leulier for bacterial strains and for consultation and advice. We thank Jonathan Landry and the EMBL sequencing core facility for RNA-sequencing and initial analysis, and Vicente Yepez for help in deeper transcriptome analysis. We would like to acknowledge the contribution of Kaida Ning in the initial analysis of the RNA-sequencing data of the fly antenna. This work was funded by the Max-Planck-Society, an EMBO young investigator grant and an ERC starting grant to IGK, and by the Juniorverbund in der Systemmedizin 'mitOmics' (FKZ 01Z × 1405A JG and VAY).

## Additional information

### Funding

| Funder | Grant reference number | Author |
| --- | --- | --- |
| Bundesministerium für Bildung und Forschung | Smartage, 01GQ1420A | Atefeh Pooryasin Stephan J Sigrist |
| Bundesministerium für Bildung und Forschung | Juniorverbund in der Systemmedizin 'mitOmics', FKZ 01Z × 1405A | Vicente A Yépez Julien Gagneur |

| | | |
|---|---|---|
| Forschungszentrum für neuro-degenerative Erkrankungen | | Stephan J Sigrist |
| H2020 European Research Council | FlyContext | Ilona C Grunwald Kadow |
| European Molecular Biology Organization | EMBO Young Investigator Small Grant | Ilona C Grunwald Kadow |
| Max-Planck-Gesellschaft | Open-access funding | Ilona C Grunwald Kadow |

The funders had no role in study design, data collection and interpretation, or the decision to submit the work for publication.

## Author contributions
Ashiq Hussain, Formal analysis, Investigation, Visualization, Methodology; Atefeh Pooryasin, Mo Zhang, Laura F Loschek, Marco La Fortezza, Habibe K Üçpunar, Formal analysis, Investigation, Visualization; Anja B Friedrich, Catherine-Marie Blais, Formal analysis, Methodology; Vicente A Yépez, Formal analysis, Investigation, Methodology; Martin Lehmann, Software, Methodology; Nicolas Gompel, Resources, Methodology, Writing—review and editing; Julien Gagneur, Software, Supervision, Investigation, Methodology, Writing—review and editing; Stephan J Sigrist, Resources, Supervision, Writing—review and editing; Ilona C Grunwald Kadow, Conceptualization, Formal analysis, Supervision, Funding acquisition, Visualization, Writing—original draft, Project administration, Writing—review and editing

## Author ORCIDs
Atefeh Pooryasin https://orcid.org/0000-0002-4025-2689
Julien Gagneur https://orcid.org/0000-0002-8924-8365
Ilona C Grunwald Kadow https://orcid.org/0000-0002-9085-4274

## Decision letter and Author response
Decision letter https://doi.org/10.7554/eLife.32018.029
Author response https://doi.org/10.7554/eLife.32018.030

## Additional files

### Supplementary files
• Source data 1. RNA sequencing all figures.
DOI: https://doi.org/10.7554/eLife.32018.022

• Supplementary file 1. Genes upregulated in brains of 7 weeks old flies vs. brains of 1 week old flies organized by GO (gene ontology) terms.
DOI: https://doi.org/10.7554/eLife.32018.023

• Supplementary file 2. Genes downregulated in brains of 7 weeks old flies vs. brains of 1 week old flies organized by GO (gene ontology) terms.
DOI: https://doi.org/10.7554/eLife.32018.024

• Transparent reporting form
DOI: https://doi.org/10.7554/eLife.32018.025

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
