## [Decision Letter]

Thank you for submitting your article "Inhibition of oxidative stress in cholinergic projection neurons rescues aging-associated olfactory circuit degeneration" for consideration by *eLife*. Your article has been favorably evaluated by K VijayRaghavan (Senior Editor) and three reviewers, one of whom, Patrik Verstreken, is a member of our Board of Reviewing Editors. The reviewers have opted to remain anonymous.

The reviewers have discussed the reviews with one another and the Reviewing Editor has drafted this decision to help you prepare a revised submission.

Summary:

Hussein et al. describe a series of experiments that identify deterioration in olfactory projection neuron function as a primary cause of aging-related behavioral and olfactory deficits. For this they use *Drosophila* and study the age and neurodegeneration-induced loss of olfactory sensitivity. They show that this phenomenon is not mainly attributable to an age-induced peripheral deficit in sensory reception, but that the function of the aged olfactory circuit (as measured by behavioral performance) is fully rescued by protecting a small population of cholinergic olfactory interneurons in the brain by scavenging oxidative stress. They then extend their approach and show that rescue is not only achieved by genetic manipulation in specific cells, but also by feeding anti-oxidants or manipulating the gut microbiome; clearly these are out of the box approaches! The study will be interesting to a broad field of researchers, as flies and mammals share not only features of architecture and function of the olfactory circuit, but also fundamental processes leading to aging and neurodegenerative diseases.

In general, the reviewers were excited about the work and appreciated that a careful examination of aging-related changes in neuron function in the fly is highly welcomed, given the importance of this model system for studies of neurodegeneration and aging.

Essential revisions:

I have summarized the essential items we felt should be addressed in a revised manuscript:

* Figure 3: labeling and/or description should be improved. It is not clear if GCaMP/Dα7-GFP is stained by an antibody here for display and for quantification. The way of quantification displayed in 3D, E and H is not explained sufficiently (what/where/how many: single boutons? large regions? single optical plane? sum of several z-planes?).

* The authors find that somata of projection neurons shrink with age. This has been described for other cell types, e.g., dopaminergic neurons. It would be nice to see, whether shrinking is specific for "vulnerable" cells, like projection neurons or dopaminergic neurons, or whether this is a general phenomenon, e.g., observed also in the olfactory sensory neurons. This can be easily quantified in the existing data displayed in Figure 2—figure supplement 1.

* The number of GH146-positive somata is found to be decreased in aged flies. We would suggest you formulate this more cautiously, i.e., reformulate more precisely. The experiments only show that fewer cell bodies express GCaMP, the authors cannot differentiate whether cells are lost or whether cells do not express the marker (anymore). Although we agree that complete loss of neurons is not a broad phenomenon in aged *Drosophila* brains, we would reformulate the arguments in the seventh paragraph of the main text more cautiously, as, for example, the ToPro staining in Figure 3 clearly indicates "holes" in the 7 week old brains.

* It would be nice (and add to the impact of the study) to show some more significantly changed genes rather than AChR and SODs. This could be done, for example, by adding the SODs to Figure 3 and creating a graph with "non-relevant" genes in Figure 4—figure supplement 6.

* The relocalization of Dα7-GFP in old flies is a very interesting finding. The connection to neurodegeneration is, however, rather vague. Here some data on number (and possibly degeneration) of branches or synaptic sites would be good to add. A degeneration of synaptic sites would likely also explain the reduced responses in the AL/number of responding boutons in the calyx (Figure 2).

* A citation (or experiment) for the efficacy of (at least) the SOD2i construct would complement this otherwise so well controlled study.

* Additional analysis of the aged GH146>SOD2 flies is maybe beyond the scope of this report. Nevertheless it raises the question whether or which of the described functional and cellular phenotypes in Figure 2 and Figure 3 are also rescued and, therefore, critical to maintain overall circuit function.

* The careful investigation of how aging influences cholinergic function, as summarized in Figure 2, is a strength, and is quite convincing. Notably, however, none of this functional work was executed in flies where genetic or environmental manipulation was shown to have reversed the phenotypic deficit at old ages. Can we be sure that the described functional deficits were equally rescued? This might be a reasonable argument for the targeted SOD expression, but it is certainly much more tenuous for the drug and microbiome experiments. Is a small or modest functional rescue sufficient to completely rescue behavior? One might argue that while the behavioral effects are very interesting, a detailed understanding of the extent to which the functional parameters are preserved are more important.

* It is well known in the aging field that lifespan and all sorts of behavioral and health-related phenotypes exhibit extreme variation among different genotypes. Indeed, the standard in the field is that all transgenic stocks be backcrossed at least 10 generations to a common background control within 6-12 months of the experiments in question. Alternatively, inducible expression systems (e.g., Geneswitch) are often used to allow comparisons between identical backgrounds with and without/drug. Apparently, this type of background control was not followed for the experiments in this manuscript. As a result, claims about lifespan differences (e.g., Figure 4—figure supplement 7) are not valid because of well-established situations in which false inference was obtained. Strain variability in cellular or neuronal functional decline is largely unknown, and to some extent, a lack of background control may have also worked a bit in the authors' favor. In mice, certain aspects of sensory decline are known to be characteristic of some mouse strains but not others (e.g., Blk6 mice go deaf quite young while other strains do not). By observing effects in "wild-type" and unrelated GAL4/UAS stocks it appears that the olfactory decline described by the authors is quite robust to strain variation. Resolving this issue is not straightforward in our opinion. We are not requesting that the authors repeat all of the experiments after backcrossing all of the genetic strains; we do believe the data in Figure 1 and Figure 2, for example. But they must address this issue in the manuscript and resolve the situations where known confounding occurs (e.g., comparing lifespans of non-backcrossed GAL4 and UAS lines).

* For the experiments involving resveratrol and microbiome manipulations, there is a critical control missing: the manipulation in question measured in young animals. For example, does short-term administration of resveratrol modify response behavior? Put another way, the proper statistical question is whether there is a significant interaction between age and treatment, which would indicate a difference in the rate of change over the lifespan. It is formally possible, for example, that resveratrol improves t-maze behavior in young animals and a similar magnitude of decline occurs from week 1 to week 7. Such a result would have a very different interpretation than that currently put forward by the authors.

* The authors show that knock-down of SOD2 levels in PN's significantly affected olfactory behavior in young flies. However, some additional experiments to fully support this finding should be performed. For instance: To what extent was SOD2 down-regulated? The RNA or protein levels should be determined. – What consequence is observed with in the neurons where SOD2 is down-regulated? Is there an overall increase in ROS levels; And when rescue/overexpression experiments with SOD2 are performed, are the ROS levels different/decreased?

---

## [Author Response]

Essential revisions:I have summarized the essential items we felt should be addressed in a revised manuscript:* Figure 3: labeling and/or description should be improved. It is not clear if GCaMP/Dα7-GFP is stained by an antibody here for display and for quantification.

We have improved the labelling and the description in the figure legend and methods. For display and for quantification, we have stained GCaMP and Dα7-GFP with an anti-GFP antibody.

The way of quantification displayed in 3D, E and H is not explained sufficiently (what/where/how many: single boutons? large regions? single optical plane? sum of several z-planes?).

We have now included a longer and more detailed description of how quantifications have been carried out in the Materials and methods section, and where appropriate also in the main text.

* The authors find that somata of projection neurons shrink with age. This has been described for other cell types, e.g., dopaminergic neurons. It would be nice to see, whether shrinking is specific for "vulnerable" cells, like projection neurons or dopaminergic neurons, or whether this is a general phenomenon, e.g., observed also in the olfactory sensory neurons. This can be easily quantified in the existing data displayed in Figure 2—figure supplement 1.

This is indeed an important point. We have followed the suggestion of the reviewers and have measured the size of the OSNs. We did not find a significant difference between the size of the cell bodies of the OSNs of young as compared to old flies. These data are included as Figure 2—figure supplement 1.

* The number of GH146-positive somata is found to be decreased in aged flies. We would suggest you formulate this more cautiously, i.e., reformulate more precisely. The experiments only show that fewer cell bodies express GCaMP, the authors cannot differentiate whether cells are lost or whether cells do not express the marker (anymore). Although we agree that complete loss of neurons is not a broad phenomenon in aged Drosophila brains, we would reformulate the arguments in the seventh paragraph of the main text more cautiously, as, for example, the ToPro staining in Figure 3 clearly indicates "holes" in the 7 week old brains.

We agree and have now reformulated this. In addition, we have included a quantification for ToPro staining, which indeed shows some reduction (included in main Figure 3). Nevertheless, other important markers such as Ncad were not reduced.

* It would be nice (and add to the impact of the study) to show some more significantly changed genes rather than AChR and SODs. This could be done, for example, by adding the SODs to Figure 3 and creating a graph with "non-relevant" genes in Figure 4—figure supplement 6.

We have followed this suggestion and have updated the graph in Figure 3. In addition, we have included two tables of genes that are up- or downregulated in aged brains.

* The relocalization of Dα7-GFP in old flies is a very interesting finding. The connection to neurodegeneration is, however, rather vague. Here some data on number (and possibly degeneration) of branches or synaptic sites would be good to add. A degeneration of synaptic sites would likely also explain the reduced responses in the AL/number of responding boutons in the calyx (Figure 2).

We thank the reviewers for this suggestion. We have included more quantifications, which have been placed as additional graphs into Figure 3. These include a quantification of an antibody staining against ChAT and a reporter construct for mitochondria. In addition, we have carried out high-resolution STED microscopy with pre- and postsynaptic markers on PN boutons. In summary, we found that the number of PN boutons in the mushroom body calyx is reduced, which explains the reduction of functional boutons found in functional calcium imaging.

* A citation (or experiment) for the efficacy of (at least) the SOD2i construct would complement this otherwise so well controlled study.

We have now added this citation as requested: “RNAi: UAS-SOD2i (Bloomington stock (BL) # 24489; (Kirby et al., 2002))”. It provides evidence of the efficacy of the used UAS-SOD2 RNAi construct. The same construct was also successfully used in Oka et al., Biogerontology 2015 and other publications.

* Additional analysis of the aged GH146>SOD2 flies is maybe beyond the scope of this report. Nevertheless it raises the question whether or which of the described functional and cellular phenotypes in Figure 2 and Figure 3 are also rescued and, therefore, critical to maintain overall circuit function.

We completely agree and we plan to address these points in the future. Unfortunately, these experiments are very time consuming and it was not possible to carry them out in the two months that were given to us by the editors. In particular, we would want to carry behavioral experiments in parallel to directly correlate phenotype in anatomy with phenotype in behaviour. Such experiments require a large number of animals (every single box in every graph represents ~500 flies or more). The underlying fly husbandry is very demanding as we need to maintain the flies until they are 7 weeks of age. At this age, a large proportion of the flies have already died meaning that we have to start with ~5 times the number of flies that are required for the actual experiment. We foresee that carrying out meaningful and well controlled functional experiments with 7 weeks old flies that overexpress SOD2 in PNs as well as in other neuron types such as OSNs will require a minimum of several months. Therefore, we hope that the reviewers will agree that this is currently beyond the scope of the present study.

We have, however, carried out anatomical analysis of flies where SOD2 was knocked-down in PNs. These flies not only show the same behavioral deficit as aged flies (as shown in the previous version of the manuscript; Figure 4), but also a very similar type and degree of anatomical and functional deficits as 7 weeks old flies. Together, these data, which have been included as new Figure 5 provide further evidence that SOD2 is indeed critical for the aging process.

* The careful investigation of how aging influences cholinergic function, as summarized in Figure 2, is a strength, and is quite convincing. Notably, however, none of this functional work was executed in flies where genetic or environmental manipulation was shown to have reversed the phenotypic deficit at old ages. Can we be sure that the described functional deficits were equally rescued? This might be a reasonable argument for the targeted SOD expression, but it is certainly much more tenuous for the drug and microbiome experiments. Is a small or modest functional rescue sufficient to completely rescue behavior? One might argue that while the behavioral effects are very interesting, a detailed understanding of the extent to which the functional parameters are preserved are more important.

We fully agree (see above), and we plan to carry out these experiments in the future, because this is a very important question. Nevertheless, we think that the current manuscript offers several interesting insights even without these additional experiments.

Finally, while we agree that it would be nice to understand the exact mechanisms of SOD2 function in aging, we would argue that the functional rescue of a behavioral deficit is the main and arguably the most important readout of a treatment. We think our data makes a strong point in showing that one localized genetic intervention can be sufficient to rescue behavioural function fully. And it provides an exciting starting point for launching further screens and functional analysis of the detailed mechanisms and potential treatments.

* It is well known in the aging field that lifespan and all sorts of behavioral and health-related phenotypes exhibit extreme variation among different genotypes. Indeed, the standard in the field is that all transgenic stocks be backcrossed at least 10 generations to a common background control within 6-12 months of the experiments in question. Alternatively, inducible expression systems (e.g., Geneswitch) are often used to allow comparisons between identical backgrounds with and without/drug. Apparently, this type of background control was not followed for the experiments in this manuscript. As a result, claims about lifespan differences (e.g., Figure 4—figure supplement 7) are not valid because of well-established situations in which false inference was obtained. Strain variability in cellular or neuronal functional decline is largely unknown, and to some extent, a lack of background control may have also worked a bit in the authors' favor. In mice, certain aspects of sensory decline are known to be characteristic of some mouse strains but not others (e.g., Blk6 mice go deaf quite young while other strains do not). By observing effects in "wild-type" and unrelated GAL4/UAS stocks it appears that the olfactory decline described by the authors is quite robust to strain variation. Resolving this issue is not straightforward in our opinion. We are not requesting that the authors repeat all of the experiments after backcrossing all of the genetic strains; we do believe the data in Figure 1 and Figure 2, for example. But they must address this issue in the manuscript and resolve the situations where known confounding occurs (e.g., comparing lifespans of non-backcrossed GAL4 and UAS lines).

We agree with these comments. Nevertheless, we would like to point out that we are not comparing lifespan between Gal4 lines, UAS lines and the combination of the two, but rather lifespan of one particular genotype to a behavioral phenotype of flies of the same genotype and therefore genetic background. The lifespan experiments were done solely to be able to correlate the lifespan of the very flies that were analysed for their behavior to their behavioral performance. The argument is that, if the flies have a very short lifespan, a strong reduction in behavioral performance of the same flies could be due to a general decline in the state of health. By contrast, we are observing relatively small effects on lifespan, but large effects in behaviour. In other words, we do not mean to imply that we are measuring lifespan to conclude how a manipulation of PNs affects lifespan. We are measuring it solely to exclude that a short or extremely long lifespan of a particular experimental or control group might explain the behavioral phenotypes. Therefore, we think these lifespan data are very important. In order to avoid misunderstandings, we have now made our motivation for including lifespan measurements clearer in the text.

* For the experiments involving resveratrol and microbiome manipulations, there is a critical control missing: the manipulation in question measured in young animals. For example, does short-term administration of resveratrol modify response behavior? Put another way, the proper statistical question is whether there is a significant interaction between age and treatment, which would indicate a difference in the rate of change over the lifespan. It is formally possible, for example, that resveratrol improves t-maze behavior in young animals and a similar magnitude of decline occurs from week 1 to week 7. Such a result would have a very different interpretation than that currently put forward by the authors.

We agree. We have included resveratrol feeding of young flies as a supplementary graph into Figure 6—figure supplement 1. Feeding resveratrol for an entire week had no positive or negative effect on the young flies’ olfactory performance.

* The authors show that knock-down of SOD2 levels in PN's significantly affected olfactory behavior in young flies. However, some additional experiments to fully support this finding should be performed. For instance: To what extent was SOD2 down-regulated?

We appreciate the concern of the reviewers. We have used a line that was previously used several times and where SOD2 levels where measured in entire embryos, larvae, pupae and adults (Kirby et al., 2002).

We have also used an alternative construct from Bloomington, which was not included in the presented results (Stock 32983 (y^1^ sc^*^ v^1^; P{TRiP.HMS00783}attP2), but produced the same behavioral phenotype. In addition, we have expressed the first construct, which we then used throughout the study (“RNAi: UAS-SOD2i (Bloomington stock (BL) # 24489; (Kirby et al., 2002))” under the control of the pan-neuronal driver elav-Gal4, which – as previously described – led to the death of the flies ~3-5 days after eclosion.

**Author response image 1. respfig1:** Figure shows olfactory preference index of 1 week old flies to 2,3-butanedione. The effect of this alternative RNAi construct of SOD2 (Stock #32983) is similar to the effect observed for the construct used throughout our study. Please note that for this experiment, the flies also carried UAS-GCaMP3.0.

**Author response image 2. respfig2:** Figure shows lifespan measurements of flies overexpressing UAS-SOD2-RNAi (Stock #24489) under the control of a Pan-neuronal driver elav-Gal4 and the Gal4/+ control. The results confirm previous results by Kirby et al., 2002.

The RNA or protein levels should be determined.

It is very difficult to carry out RNA or protein measurements selectively on PNs. We do not have an antibody against SOD2 that functions in wholemount immunohistochemistry. To extract RNA just from these neurons is very challenging. Given that two independent lines showed the same behavioral phenotype and both lines have been used previously, we hope that the reviewers will agree that a proof beyond doubt that SOD2 is truly downregulated specifically in PNs is beyond the scope of the study.

What consequence is observed with in the neurons where SOD2 is down-regulated? Is there an overall increase in ROS levels; And when rescue/overexpression experiments with SOD2 are performed, are the ROS levels different/decreased?

We have analysed the consequence of SOD2 knockdown on PN morphology and function. We find that, while SOD2 knockdown does not reduce the number of PNs significantly, it has a significant impact on the size of PN cell bodies. These are reduced resembling the phenotype observed in PNs of old flies.

Furthermore, knockdown of SOD2 in PNs had a strong effect on the responses of PNs in calcium imaging. Similar to what we found in brains of old flies, the response per bouton is not reduced, but the number of responsive boutons is strongly diminished. These data have been included in a new Figure 5.

Regarding the ROS stainings, Oka et al., 2015 have used a reporter gene GstD1-GFP expressed under the control of elav-Gal4 to monitor ROS in all neurons in the brain. Their results demonstrate that knock-down of SOD2 using the same construct from Kirby et al. as used in this study increases ROS in the whole brain.

We have previously attempted to carry out ROS stainings using Dihydroethidium (DHE) on old brains with little success. We have followed the protocol as described in Owusu-Ansah et al. in Protocol Exchange. While we did see staining in old brains, we also observed stainings in younger brains occasionally. By doing different kinds of controls (staining procedure, person who dissected and stained, movements such as shaking etc.), we finally came to the conclusion that the staining is extremely sensitive to all kinds of perturbations. In particular, we found that confocal scanning increased the staining, possibly because of the effect of the strong laser light on the living tissue. We, therefore, concluded that this staining is too unreliable in our hands to analyse neuron-specific effects, and discontinued our efforts in this direction.

In light of the reviewers’ comments, we aimed at analysing ROS upon SOD2 knockdown in PNs in 1 week old brains. Unfortunately, this led to the same situation and we, once again, do not feel comfortable to conclude anything from these results, because of the variable stainings. While we do not doubt that this procedure works in other labs, we have not been able to develop a staining and imaging protocol that reliably produces consistent results independent of the experimenter, way of dissection, way of staining etc.